# Association of Late Radiographic Assessment of Lung Edema Score with Clinical Outcome in Patients with Influenza-Associated Acute Respiratory Distress Syndrome

**DOI:** 10.3390/diagnostics13233572

**Published:** 2023-11-30

**Authors:** Hsiao-Chin Shen, Chun-Chia Chen, Wei-Chih Chen, Wen-Kuang Yu, Kuang-Yao Yang, Yuh-Min Chen

**Affiliations:** 1Department of Chest Medicine, Taipei Veterans General Hospital, Taipei 112, Taiwan; joshua92632@gmail.com (H.-C.S.);; 2Department of Medical Education, Taipei Veterans General Hospital, Taipei 112, Taiwan; 3Faculty of Medicine, School of Medicine, College of Medicine, National Yang Ming Chiao Tung University, Taipei 112, Taiwan; 4Institute of Emergency and Critical Care Medicine, National Yang Ming Chiao Tung University, Taipei 112, Taiwan; 5Cancer Progression Research Center, National Yang Ming Chiao Tung University, Taipei 112, Taiwan

**Keywords:** influenza, Radiographic Assessment of Lung Edema score, acute respiratory distress syndrome, chest X-ray

## Abstract

**Background:** Influenza virus infection leads to acute pulmonary injury and acute respiratory distress syndrome (ARDS). The Radiographic Assessment of Lung Edema (RALE) score has been proposed as a reliable tool for the evaluation of the opacity of chest X-rays (CXRs). This study aimed to examine the RALE scores and outcomes in patients with influenza-associated ARDS. **Methods:** Patients who were newly diagnosed with influenza-associated ARDS from December 2015 to March 2016 were enrolled. Two independent reviewers scored the CXRs obtained on the day of ICU admission and on days 2 and 7 after intensive care unit (ICU) admission. **Results:** During the study, 47 patients had influenza-associated ARDS. Five died within 7 days of ICU admission. Of the remaining 42, non-survivors (*N* = 12) had higher Sequential Organ Failure Assessment scores (SOFA) at ICU admission and higher day 7 RALE scores than survivors (*N* = 30). The day 7 RALE score independently related to late in-hospital mortality (aOR = 1.121, 95% CI: 1.014–1.240, *p* = 0.025). **Conclusions:** The RALE score for the evaluation of opacity on CXRs is a highly reproducible tool. Moreover, RALE score on day 7 was an independent predictor of late in-hospital mortality in patients with influenza-associated ARDS.

## 1. Introduction

Influenza, a communicable ailment, is brought on by the invasion of either the influenza A virus or influenza B virus. Additionally, there are known subtypes, influenza C virus and influenza D virus [1]. It spreads rapidly, particularly during the winter season, and it is prevalent worldwide. Nearly annual influenza epidemics result in global deaths of more than 500,000 per year [2]. The symptoms associated with being infected with the influenza virus can vary greatly, ranging from mild symptoms of upper respiratory tract infection to severe cases of pneumonia and acute respiratory distress syndrome (ARDS), which can be life-threatening. These severe outcomes may result from the influenza virus infection itself or from secondary infections caused by bacteria, other viruses, or fungi [3,4,5,6].

ARDS is a diffuse pulmonary alveolar and endothelial injury secondary to the inflammatory process. This syndrome manifests as acute hypoxemia and requires mechanical ventilation and oxygen for life support [7]. Noncardiogenic pulmonary edema is a critical characteristic of both the pathogenesis and prognosis of ARDS [8], but the current methods for quantifying the severity of pulmonary edema are invasive (e.g., right cardiac catheterization) or time consuming (e.g., computed tomographic imaging). The Radiographic Assessment of Lung Edema (RALE) score is a proposed method for evaluating the density and extent of opacities on chest radiographs in patients diagnosed with ARDS; it can be utilized to assess both the degree of pulmonary edema and the severity of ARDS [9,10,11,12,13,14]. Jabaudon et al. reported that changes in the RALE score during the first 3 days after admission to the intensive care unit (ICU) are linked to the survival rate of patients who have ARDS [10]. Nevertheless, the relationship between the RALE score at a later stage and the in-hospital mortality rate of patients with influenza-induced ARDS has yet to be established.

The objective of this retrospective study was to investigate the associations of serial RALE scores and clinical outcomes with the diagnosis of influenza-associated ARDS. It also aimed to determine the best RALE scores across different lengths of ICU stay for determining late in-hospital mortality.

## 2. Materials and Methods

### 2.1. Study Design and Patient Selection

This retrospective observational study was conducted at a 3000-bed comprehensive tertiary medical center. The Institutional Ethical Review Board of the hospital approved the study. The IRB granted a waiver for informed consent owing to the retrospective nature of the observational study. The study identified patients who received a new diagnosis of influenza-associated ARDS between December 2015 and March 2016. The following are the exclusion criteria: (1) a diagnosis of virology-proven influenza that was not confirmed by either rapid influenza diagnostic test or reverse-transcription polymerase chain reaction (RT-PCR), (2) age < 18 years, (3) death within 7 days after ICU admission, and (4) incomplete data. The study participants were segregated into two distinct groups for analysis. The first group comprised patients who were discharged alive from the hospital, also named as the survival group. The second group consisted of patients who died during hospitalization, referred to as the nonsurvival group.

### 2.2. Definitions of ARDS and RALE Score

Throughout the study period, ARDS evaluation was conducted following the Berlin definition, outlined as: (1) acute onset of respiratory distress within one week, (2) the presence of opacities in bilateral lungs as confirmed by imaging studies, which cannot be entirely attributed to cardiogenic pulmonary edema, and (3) hypoxemia identified by the ratio of arterial oxygen partial pressure to the fraction of inspired oxygen (PaO_2_/FiO_2_) equal to or less than 300, when applying positive end-expiratory pressure or continuous positive airway pressure of 5 cmH_2_O or greater. The details of the RALE score have been described in previous publications [9,15,16] and are illustrated in Figure A1. To determine the RALE score, the radiograph’s consolidation and density scores were assessed for each quadrant (i.e., upper/lower right and upper/lower left quadrants) and were added together. The consolidation scores were determined by evaluating the degree of opacities in each quadrant and assigned scores based on the extent of involvement. This entailed assigning scores to each quadrant based on the extent of consolidation and density observed, with the following criteria: none (0 points), <25% (1 point), 25–50% (2 points), 50–75% (3 points), and >75% (4 points). Additionally, the opacity density was assessed for each quadrant and given a score of hazy (1 point), moderate (2 points), or dense (3 points). Two independent pulmonologists (HCS and CCC) scored the chest X-rays (CXRs) obtained on the day of ICU admission (day 0) and on days 2 and 7 following ICU admission.

### 2.3. Data Collection and Severity Evaluation

Data on CXRs, demographic characteristics, and preexisting comorbidities were extracted from both medical charts and electronic medical records. Disease severity was assessed using several scoring systems, including the Acute Physiology and Chronic Health Evaluation (APACHE) II score, Sequential Organ Failure Assessment (SOFA) score, Pneumonia Severity Index (PSI), and CURB-65 score, all of which were determined on the day of admission to ICU. The results of the laboratory tests at ICU admission, medications including osteltamivir, corticosteroid, vasopressors, and sedative agents, and the application of renal replacement therapy and extracorporeal membrane oxygenation (ECMO) support were also obtained and analyzed.

### 2.4. Outcome Evaluation

The present study assessed several outcomes, including the duration of ICU and hospital stays, mechanical ventilation, and all-cause mortality rate at the time of discharge. Interobserver agreement was also assessed for the RALE score calculation. All patients were monitored from admission to either death or discharge.

### 2.5. Statistical Analysis

The reliability of the RALE scores between independent reviewers was evaluated by computing the average-measures intraclass correlation coefficient (ICC) using a two-way random consistency model for the scores on day 0 (i.e., ICU admission day), as well as for days 2 and 7 after ICU admission. The agreement between independent reviewers was visualized using Bland–Altman plots. The Mann–Whitney U test was used to analyze continuous variables, which were presented as median values along with the interquartile range (IQR). On the other hand, categorical variables were analyzed using the chi-square test and reported as counts and percentages. We conducted a multivariate logistic regression analysis to identify independent factors related to in-hospital mortality, considering variables with a *p*-value less than 0.2. The resulting odds ratios (ORs) with 95% confidence intervals (CIs) were calculated. A significance level of *p* < 0.05 (two-tailed) was used for all statistical analyses. The software packages used for data analysis were SPSS Statistics for Windows/Macintosh version 25.0 (IBM, Armonk, NY, USA) and MedCalc version 20.215.

## 3. Results

### 3.1. Patient Characteristics

Fifty patients were admitted with the diagnosis of influenza-associated ARDS. Based on the exclusion criteria, a total of eight patients were ineligible and therefore excluded from the study (Figure 1). The 42 remaining patients with a median age of 65 years (IQR: 55–79) were enrolled. Among the 42 patients enrolled in the current study, influenza was diagnosed in three individuals via bronchoalveolar lavage, two through sputum analysis, while the remaining cases were identified using nasopharyngeal specimens. As a whole group, the durations of mechanical ventilation, ICU stay, and hospitalization were 17 (IQR: 11–31), 19 (IQR: 12–26), and 28 (IQR: 17–51) days, respectively. At ICU admission, the median SOFA and APACHE II scores were 7 (IQR: 5–8) and 23 (IQR: 16–30), respectively. During ICU admission, two (4.8%) patients received new renal replacement therapy, while 14 (33.3%) of the patients received ECMO support.

### 3.2. Demographic and Baseline Clinical Data

Table 1 presents the demographic characteristics and baseline clinical variables for the 42 enrolled patients. Of these patients, 30 (71.4%) survived and were subsequently discharged from the hospital. Notably, the nonsurvival group had a higher SOFA score on admission compared to the survival group (*p* = 0.049). However, no significant differences were observed between the two groups regarding demographics and other baseline clinical variables.

### 3.3. Interobserver Agreement for RALE Scores

The RALE scores of the two observers were compared, and the results are shown in Figure A2. The ICCs for RALE scores were excellent at ICU admission (ICC = 0.872, 95% CI = 0.781–0.927), on day 2 (ICC = 0.884, 95% CI = 0.798–0.934), and on day 7 after ICU admission (ICC = 0.908, 95% CI = 0.834–0.950), indicating a high degree of agreement between the observers (Figure A2A–C). Bland–Altman plots also showed a strong agreement across the range of RALE scores (Figure A2D–F).

### 3.4. Clinical Complications and Hospital Outcomes

Table 2 compares the clinical complications and hospital outcomes between the two groups. The nonsurvival group demonstrated a higher incidence of bacteremia among the patients (*p* = 0.026) and vasopressor use (*p* = 0.014) than the patients in the survival group. However, no significant differences in the duration of invasive mechanical ventilation (*p* = 0.503), length of ICU stay (*p* = 0.062), and length of hospitalization (*p* = 0.944) were observed between these two groups.

### 3.5. RALE Score and Survival

Figure 2 shows the comparisons of RALE scores on days 0, 2, and 7. No significant differences in RALE scores were observed at ICU admission (*p* = 0.103, Figure 2A) and on day 2 after ICU admission (*p* = 0.078, Figure 2B). However, compared with the survival group, the nonsurvival group had higher RALE score at day 7 after ICU admission (*p* = 0.01).

### 3.6. Independent Predicting Factor for In-Hospital Mortality

Variables associated with in-hospital mortality (*p* < 0.2) were subjected to multivariate logistic regression analysis to investigate whether the RALE score is an independent predictor for in-hospital mortality. The results after adjustments for the SOFA score, APACHEII score, PSI at admission, vasopressors use, and bacteremia development revealed that the RALE scores at days 0 and 2 were not associated with in-hospital mortality (Table A1 and Table A2). However, the RALE scores at day 7 are an independent predictor for in-hospital mortality (aOR = 1.121, 95% CI: 1.014–1.240, *p* = 0.025, Table 3).

## 4. Discussion

Influenza-associated ARDS is a complex and severe disease that requires critically supportive care in the ICU. Considering the findings of the current study, the RALE score is reproducible. Furthermore, higher RALE scores at day 7 after ICU admission were independently associated with in-hospital mortality.

The CXR was systematically scored to quantify the severity of pulmonary edema and to obtain a RALE score [9]. Our study verified the reproducibility of the RALE score among diverse patient cohorts and independent reviewers, as evidenced by the high intraclass correlation coefficient (ICC) obtained, as well as in the original study that introduced the RALE score [9]. The RALE scoring system is dependent on physician interpretation of standard chest radiographs, which enhances its practicality and ease of use as a tool for quantifying the extent of pulmonary edema. This radiographic scoring has been found to be associated with the severity and clinical outcomes of ARDS [9,10,11,12]. However, an association was not observed between the RALE scores of the baseline (at ICU admission) CXR and in-hospital mortality. It is possible that the initial radiographic evaluation of pulmonary edema may not be sufficient to capture extrapulmonary organ damage in patients with ARDS. Studies that assessed the RALE score in patients with ARDS from causes other than influenza had conflicting results in terms of the prognostic capacity of the baseline RALE score [10,11,17]. This finding supports the absence of association between the baseline RALE score and in-hospital mortality [11,17].

The RALE scores on day 7 after ICU admission were independently associated with in-hospital mortality, supporting previous findings that the dynamic change in RALE score rather than the baseline RALE has an association with morality [11,12]. This phenomenon can also be attributed to the causes of influenza-associated ARDS. For the pathogenesis of influenza-associated ARDS, influenza virus primarily targets epithelial cells, and because of viral infection, epithelial cells may produce cytokines that stimulate the recruitment of leukocytes and activation of adjacent endothelial cells. The activation of endothelial cells and infiltration of leucocytes further exacerbate inflammation, contributing to the development of ARDS. Additionally, influenza infection may occur in conjunction with or be followed by secondary bacterial infection, which can also trigger ARDS [18,19]. The RALE scores on day 7 after ICU admission may have a better ability to capture secondary bacterial pneumonia than baseline RALE score, resulting in a stronger association with in-hospital mortality.

Past experiences, such as the outbreaks of severe acute respiratory syndrome [20], middle east respiratory syndrome [21], and coronavirus disease 2019 [22], have demonstrated that clusters of viral pneumonia that manifest within a brief period can serve as a critical signal of an outbreak or pandemic. Therefore, there is a pressing need for a fast, precise, and economical method for identifying viral pneumonia. Chest X-ray (CXR) is an effective tool for the prompt detection of outbreaks caused by novel viruses [23]. The favorable findings in influenza present that the RALE score may be a potential tool for the outcome prediction of viral pneumonia.

Our study had several strengths. First, patients recruited in the current study all underwent chest X-rays on the designated study day, allowing for the calculation of the RALE score. Secondly, the RALE score was shown to be both feasible and reliable, with low variability observed between the scorers. Thirdly, pulmonologists are able to calculate the RALE score at bedside without invasive procedures, and this invaluable information can assist pulmonologists in promptly treating ARDS patients. Finally, to our knowledge, this study was the first to investigate the association of RALE score with the severity and prognosis of influenza-associated ARDS. Several limitations must be acknowledged in this study. First, this study is a single-center study with a small sample size, in which only 42 cases were included in the final analysis. Second, this study is a retrospective cohort study, which may involve some missing medical data. Third, not all patients with influenza were diagnosed via RT-PCR. Some patients were enrolled via rapid influenza diagnostic test, which may cause false-positive results. Finally, our study focused solely on patients diagnosed with influenza-induced ARDS who required ICU admission. Therefore, the generalizability of our findings to patients with less severe cases of influenza remains uncertain. For better understanding of the relationship of RALE score and influenza-associated ARDS, a well-designed prospective clinical study is necessary.

## 5. Conclusions

In conclusion, the RALE score offers a reliable means of evaluating the degree of radiographic edema and is easily implemented in clinical practice. The results show that the RALE score of baseline CXR was not associated with in-hospital mortality. However, the RALE scores on day 7 after ICU admission were an independent predictor of in-hospital mortality.

## Figures and Tables

**Figure 1 diagnostics-13-03572-f001:**
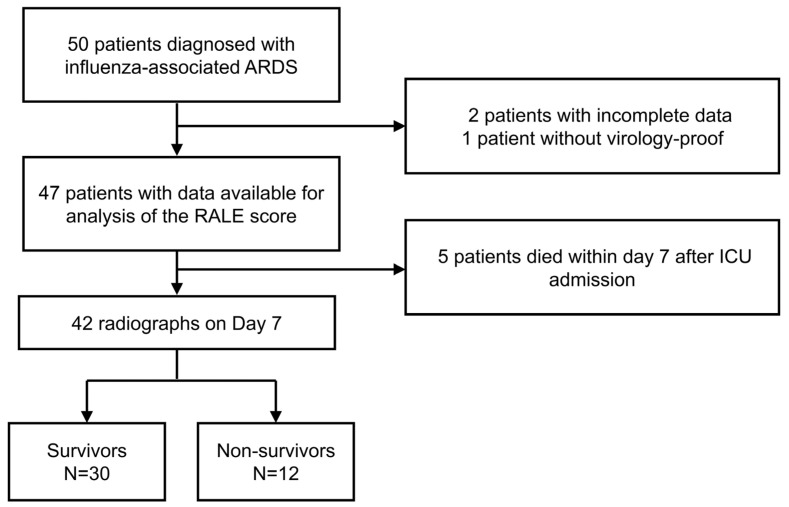
Flow of patients with ARDS who were included in the RALE score analysis.

**Figure 2 diagnostics-13-03572-f002:**
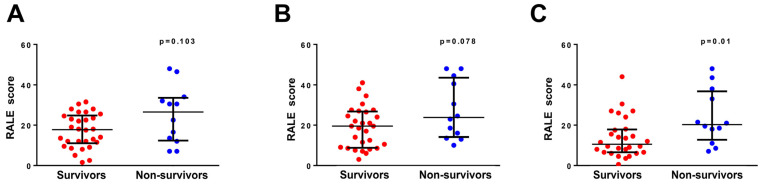
RALE score and survival status. RALE score at ICU admission (**A**), on day 2 after ICU admission (**B**), and on day 7 after ICU admission (**C**) in the survival and nonsurvival groups. The nonsurvival group has a higher RALE score than the survival group on day 7 after ICU admission *(p* = 0.010).

**Table 1 diagnostics-13-03572-t001:** Demographic characteristics and baseline clinical data of the two study groups.

	Survivors (*n* = 30)	Non-Survivors (*n* = 12)	*p* Value
Age (years)	65 [55–80]	65 [56–81]	0.933
Male sex	22 (73)	10 (83)	0.492
BMI (kg/m^2^)	23 [21–27]	24 [21–27]	0.933
**Comorbidities**			
Diabetes mellitus	12 (40)	5 (42)	0.921
Cardiovascular disease	19 (63)	6 (50)	0.426
Chronic kidney disease	4 (13)	1 (8)	0.651
**At ICU admission**			
WBC (10^3^/mm^3^)	8.4 [6.9–14.0]	7.6 [5.1–11.5]	0.411
Albumin (g/dL)	2.9 [2.4–3.1]	2.6 [2.4–2.8]	0.228
CRP (mg/dL)	21.4 [6.3–30.0]	12.3 [8.2–15.0]	0.344
Lactate (mg/dL)	13.8 [8.2–19.8]	14.7 [10.6–20.4]	0.616
Creatine kinase (U/L)	144.0 [67.0–484.5]	186.5 [49.0–313.0]	0.954
ALT (IU/L)	28.5 [17.4–40.3]	42.0 [20.3–90.8]	0.200
**Severity**			
CURB-65	3.0 [1.0–4.0]	2.5 [2.0–3.8]	0.830
PSI	114.5 [97.0–148.8]	126.0 [124.0–144.8]	0.140
SOFA	7.0 [4.0–8.0]	7.5 [6.3–12.8]	0.049
APACHEII	20.5 [15.0–29.3]	25.5 [20.5–30.0]	0.112
**RALE score**			
Day 0	17.8 [11.0–24.8]	26.5 [12.4–33.5]	0.103
Day 2	19.5 [8.8–26.8]	23.8 [14.1–43.5]	0.078
Day 7	9.5 [6.3–17.8]	20.3 [12.8–36.8]	0.010

BMI, body mass index; WBC, white blood cell; CRP, C-reactive protein; ALT, alanine aminotransferase; PSI, Pneumonia Severity Index; APCHEII, Acute Physiology and Chronic Health Evaluation II; SOFA, Sequential Organ Failure Assessment score; ICU, intensive care unit; Day 0, at ICU admission. Continuous data are expressed as median with interquartile range [IQR] and are compared by Mann–Whitney U test. Categorical variables are expressed as number of patients (%) and are compared by chi-square test. A *p* value of <0.05 is considered statistically significant.

**Table 2 diagnostics-13-03572-t002:** Comparisons of clinical complications, treatment, and hospital outcomes between the two study groups.

	Survivors (*n* = 30)	Non-Survivors (*n* = 12)	*p* Value
**Complications**			
Bacteremia	2 (7)	1 (33)	0.026
**Treatments**			
IMV	27 (90)	12 (100)	0.256
Corticosteroid use	16 (53)	8 (67)	0.430
Vasopressors use	10 (33)	9 (75)	0.014
**Outcomes**			
ICU days	18 [9–22]	26 [16–38]	0.062
Hospitalization days	28 [16–55]	30 [18–48]	0.944
IMV days	16 [11–31]	18 [12–38]	0.503

HAP, hospital acquired pneumonia; IMV, invasive mechanical ventilation; NIPPV, noninvasive positive pressure ventilators; ICU, intensive care unit. Continuous data were expressed as median with interquartile range [IQR] and were compared by Mann–Whitney U test. Categorical variables were expressed as number of patients (%) and were compared by chi-square test. A *p* value of <0.05 was considered statistically significant.

**Table 3 diagnostics-13-03572-t003:** Multivariate logistic regression analyses of Day 7 RALE and other factors associated with death.

	aOR (95% CI)	*p* Value
Initial SOFA	1.068 (0.795–1.435)	0.660
Vasopressors	1.606 (0.205–12.573)	0.652
Bacteremia	9.669 (0.587–159.133)	0.112
APACHE II	0.974 (0.831–1.143)	0.750
PSI	1.027 (0.991–1.065)	0.144
Day 7 RALE	1.121 (1.014–1.240)	0.025

Initial SOFA: Sequential Organ Failure Assessment Score at the first day of ICU admission; APCHEII, Acute Physiology and Chronic Health Evaluation II; PSI, Pneumonia Severity Index; Day 7, at Day 7 after ICU admission; OR, odds ratio; CI, confidence interval. aOR, odds ratio after adjustment for other confounding factors. A *p* value of <0.05 was considered statistically significant.

## Data Availability

The datasets used and/or analyzed during the current study available from the corresponding author on reasonable request.

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
