# Peer review of "Association of Late Radiographic Assessment of Lung Edema Score with Clinical Outcome in Patients with Influenza-Associated Acute Respiratory Distress Syndrome"

_diagnostics, 2023, doi:10.3390/diagnostics13233572_

Round 1

Reviewer 1 Report

Comments and Suggestions for Authors In the current study, Shen and colleagues explored the probable interconnection of influenza A virus-driven ARDS-associated death of patients with changes identified in chest X-rays (CXRs). The authors found that a high radiographic assessment of lung edema (RALE) score was significantly associated with poor prognosis in these patients. The findings can be used to expedite the diagnosis of severe ARDS associated with influenza A virus infection by allowing the use of CXRs data instead of more time-consuming computed tomography. This paper can be accepted for publication in Diagnostics after correction of minor comments: 1. Please give transcripts of ICU (intensive care unit) and SOFA (Sequential Organ Failure Assessment) in the Abstract. 2. Line 34: Please put spaces in influenzavirus C and influenzavirus D. 3. Line 53: What does ARDS.8 mean? If it is a reference, please format it according to journal rules. 4. Line 81: What sentence does "As described in previous publications [7, 13, 14]" refer to? Please correct it. 5. Lines 175-176: p = 0.001 and p = 0.01 in Figure 2C. Which is the correct variant?

Author Response

Responses to Comments from Reviewers (Diagnostics-2715911)

We would like to thank the reviewers for their extensive assessment of our manuscript, and for important and helpful comments and suggestions. We have taken all the remarks into account, in a manner that is described in detail below together with our answers to certain comments. We have responded to all the reviewer’s comments in a point-by-point fashion and have revised the manuscript accordingly. The revised portions are indicated by red font. We think that, following the reviewers’ suggestions, our manuscript has gained in clarity and hope that the changes made will be considered satisfactory.

Responses to Comments from Reviewer #1:

Comment 1. Please give transcripts of ICU (intensive care unit) and SOFA (Sequential Organ Failure Assessment) in the Abstract.

Response: We thank the reviewer for this excellent suggestion. In response to the suggestion, we have made added the transcripts of ICU (intensive care unit, line 20) and SOFA (Sequential Organ Failure Assessment, line 22 and23) to the abstract.

Comment 2. Line 34: Please put spaces in influenzavirus C and influenzavirus D.

Response: We apologize for the error we made which has been corrected (lines 34 and 35).

Comment 3. Line 53: What does ARDS.8 mean? If it is a reference, please format it according to journal rules.

Response: We apologize for the error we made, which has been corrected (line 54).

Comment 4. Line 81: What sentence does "As described in previous publications [7, 13, 14]" refer to? Please correct it.

Response: We apologize for the errors in the sentence and references, which have been corrected (line 82 and 83).

Comment 5. Lines 175-176: p = 0.001 and p = 0.01 in Figure 2C. Which is the correct variant?

Response: We apologize for the error we made which has been corrected (p = 0.01, line 180 and 181).

Reviewer 2 Report

Comments and Suggestions for Authors

Dear Authors,

Thank you very much for your manuscript dealing with an important medical issue. Please pay attention to the following comments and queries, pertaining to your manuscript:

1.      Line 26. Please correct as such: RALE score

2.      Line 37. I suppose you mean that 500,000 fatalities are annually observed worldwide. Please make this point clear.

3.      Lines 40-41. Please correct as such and provide appropriate references: which can be life-threatening, and may occur either due to the influenza virus infection itself or due to superinfection caused by bacteria, other viruses or fungi.

4.      Line 53. Please correct as such: ARDS [8]. Nevertheless, etc.

5.      Line 67. Was rapid influenza diagnostic test or RT-PCR conducted in sputum or in BAL?

6.      Line 70. Please correct as such: also named as the survival group.

7.      Line 90. Please explain why the evaluation of chest X-rays was only conducted by pulmonologists and not also from radiologists and provide some information about the professional experience of pulmonologists regarding the evaluation of radiological examinations.

8.      Line 138. Please correct as such: had a higher SOFA score on admission compared to the survival group (p = 0.049).

9.      Line 151. Please correct as such: a high degree of agreement between the observers.

10.   Please rather use the term: in-hospital mortality.

11.   Line 235. “First, no patient was lost to follow up”. There wasn’t actually any follow-up in your study, as the patients who died within 7 days after ICU admission were excluded from your retrospective study. Please avoid this point.

Best Regards

Comments on the Quality of English Language

Extensive editing of English language required

Author Response

Responses to Comments from Reviewer 2 (Diagnostics-2715911)

We would like to thank the reviewers for their extensive assessment of our manuscript, and for important and helpful comments and suggestions. We have taken all the remarks into account, in a manner that is described in detail below together with our answers to certain comments. We have responded to all the reviewer’s comments in a point-by-point fashion and have revised the manuscript accordingly. The revised portions are indicated by red font. We think that, following the reviewers’ suggestions, our manuscript has gained in clarity and hope that the changes made will be considered satisfactory. Thank you for pointing out the areas where our article's English needed improvement. This year, we have already had the article edited by an English language editor (proof of editing is attached as following). We have also made every effort to correct the issues in the article that you highlighted.

Responses to Comments from Reviewer #2:

Comment 1. Line 26. Please correct as such: RALE score

Response: We apologize for the error we made which has been corrected (line 27).

Comment 2. Line 37. I suppose you mean that 500,000 fatalities are annually observed worldwide. Please make this point clear.

Response: We thank the reviewer for reminding us regarding this important issue. Actually, nearly annual influenza epidemics result in a global death more than 500,000 per year. In response to the suggestion, we have added to the introduction (line 36 and 37).

Comment 3. Lines 40-41. Please correct as such and provide appropriate references: which can be life-threatening, and may occur either due to the influenza virus infection itself or due to superinfection caused by bacteria, other viruses or fungi.

Response: We thank the reviewer for this constructive suggestion. In response to the suggestion, we have added the statements and references (line 40-42, and reference 5 and 6).

Comment 4. Line 53. Please correct as such: ARDS [8]. Nevertheless, etc.

Response: We apologize for the error we made which has been corrected (line 54).

Comment 5. Line 67. Was rapid influenza diagnostic test or RT-PCR conducted in sputum or in BAL?

Response: We thank the reviewer for reminding us regarding this important issue. Among the 42 patients enrolled in our study, influenza was diagnosed in 3 individuals via bronchoalveolar lavage (BAL), 2 through sputum analysis, while the remaining cases were identified using nasopharyngeal specimens. Typically, when symptoms first manifest, most patients undergo an initial nasopharyngeal specimen test due to its rapidity. If the nasopharyngeal test yields a positive result, our hospital generally doesn’t proceed with further lower respiratory tract tests. However, for hospitalized patients experiencing respiratory failure and testing negative on upper respiratory tract specimens, we collected lower respiratory tract samples in adherence to the 2018 IDSA guideline for seasonal influenza management [1]. Additionally, a previous study indicated that nasopharyngeal swabs possess a sensitivity comparable to sputum in detecting influenza [2]. Therefore, employing nasopharyngeal specimens for initial influenza screening remains appropriate, even in critically ill patients. In response to reviewer comments, we have added statement to the result (line 129-131).

Comment 6. Line 70. Please correct as such: also named as the survival group.

Response: We apologize for the error we made which has been corrected (line 72 and 73).

Comment 7. Line 90. Please explain why the evaluation of chest X-rays was only conducted by pulmonologists and not also from radiologists and provide some information about the professional experience of pulmonologists regarding the evaluation of radiological examinations.

Response: We thank the reviewer for reminding us regarding this important issue. Firstly, the RALE score was developed by pulmonologists for their familiarity with the definition of ARDS, which was not solely relied on chest radiographs [3]. Secondly, pulmonologists are able to calculate the RALE score at bedside without invasive procedures, and this invaluable information could assist pulmonologists in promptly treating ARDS patients. In response to review suggestion, we had added statement to the discussion (line 243-245).

Comment 8. Line 138. Please correct as such: had a higher SOFA score on admission compared to the survival group (p = 0.049).

Response: We apologize for the error we made which has been corrected (line 142 and 143).

Comment 9. Line 151. Please correct as such: a high degree of agreement between the observers.

Response: We apologize for the error we made which has been corrected (line 156 and 157).

Comment 10. Please rather use the term: in-hospital mortality.

Response: Thanks for your constructive suggestions. In response to the suggestion, we have replaced the term “mortality” to “in-hospital mortality” in the revised manuscript (line 24, 27, 55, 60, 118, 186, 187, 189, 191, 192, 203, 212, 217-219, 230, 231, and 261-262).

Comment 11. Line 235. “First, no patient was lost to follow up”. There wasn’t actually any follow-up in your study, as the patients who died within 7 days after ICU admission were excluded from your retrospective study. Please avoid this point.

Response: Thanks for your excellent suggestion. The strength of the study was that patients recruited in the study all underwent chest X-rays on the designated study day, allowing for the calculation of the RALE score. In response to the suggestion, we have deleted the sentence and added statement to the discussion (line 240-242).

Reference

  1. Uyeki TM, Bernstein HH, Bradley JS, Englund JA, File TM, Fry AM, Gravenstein S, Hayden FG, Harper SA, Hirshon JM, Ison MG, Johnston BL, Knight SL, McGeer A, Riley LE, Wolfe CR, Alexander PE, Pavia AT. Clinical practice guidelines by the Infectious Diseases Society of America: 2018 Update on diagnosis, treatment, chemoprophylaxis, and institutional outbreak management of seasonal Influenzaa. Clinical Infectious Diseases. 2018. 68(6):e1-e47.
  2. Covalciuc KA, Webb KH, Carlson CA. Comparison of four clinical specimen types for detection of influenza A and B viruses by optical immunoassay (FLU OIA test) and cell culture methods. J Clin Microbiol. 1999;37(12):3971-3974.
  3. Warren MA, Zhao Z, Koyama T, Bastarache JA, Shaver CM, Semler MW, Rice TW, Matthay MA, Calfee CS, Ware LB. Severity scoring of lung oedema on the chest radiograph is associated with clinical outcomes in ARDS. Thorax. 2018;73(9):840-846.

Round 2

Reviewer 2 Report

Comments and Suggestions for Authors

Dear Authors,

thank you for providing comprehensive and convincing answers to the questions and queries expressed by me and the other Reviewers and made changes, which have contributed to the optimization of your manuscript and increased the publishing potential of your work.

Best Regards